# MoleBridge: Synthetic Space Projecting with Discrete Markov Bridges

**Rongchao Zhang**[1], **Yu Huang**[2],[*] **Yongzhi Cao**[1], **Hanpin Wang**[1]
[1]Key Laboratory of High Confidence Software Technologies (Peking University),
Ministry of Education, School of Computer Science, Peking University
[2]National Engineering Research Center for Software Engineering, Peking University
rczpku@163.com, hy@pku.edu.cn

## Abstract

Molecular synthetic space projecting is a critical technique in *de novo* molecular design, which aims to rectify molecules without synthesizability guarantee by converting them into synthetic postfix notations. However, the vast synthesizable chemical space and the discrete data modalities involved pose significant challenges to postfix notation conversion benchmarking. In this paper, we exploit conditional probability transitions in discrete state space and introduce MoleBridge, a deep generative model built on the Markov bridge approach for designing postfix notations of molecular synthesis pathways. MoleBridge consists of two iterative optimizations: i) Autoregressive extending of notation tokens from molecular graphs, and ii) generation of discrete reaction postfix notations through Markov bridge, where noisy token blocks are progressively denoised over multi-step iterations. For the challenging second iteration, which demands sensitivity to incorrect generative probability paths within intricate chemical spaces, we employ a thinking and denoising separation approach to denoise. Empirically, we find that MoleBridge is capable of accurately predicting synthesis pathways while exhibiting excellent performance in a variety of application scenarios.

## 1 Introduction

*De novo* molecular design has garnered considerable attention across various research domains in life sciences [77, 63, 2]. Among these developments, the majority of cutting-edge breakthroughs are driven by deep generative approaches [48, 42, 21]. With the promise comes a challenge: unlike traditional combinatorial optimization approaches [59, 9] constrained by virtual libraries, generative models typically generate structures that lie outside the synthesizable chemical space [18]. Of these, only a vanishing small percentage will be experimentally realizable. Recently, the immense potential of projecting synthetic space for the rectification of non-synthesizable molecules has given rise to a milestone paradigm [46], where desired synthesizable molecules [31] from structurally similar analogs are now made available. This paradigm is centered on generating synthetic pathways from purchasable chemical building blocks and deriving designed molecules in postfix notations, which can rival chemically expert-defined rules.

Despite their promising advancements, there are still significant gaps to fill before generative models become practical for synthesizable pathway design. i) First, chemical space theory predicts that the number of compounds synthesizable by humans could reach $10^{63}$—an enormous space [5, 27]. For this purpose, it is essential that the model can explore, on a sequence-to-sequence basis, synthetic pathways of arbitrary length. ii) Second, unlike natural language, molecular postfix notation sequences

---

[*]Corresponding Author.

39th Conference on Neural Information Processing Systems (NeurIPS 2025).

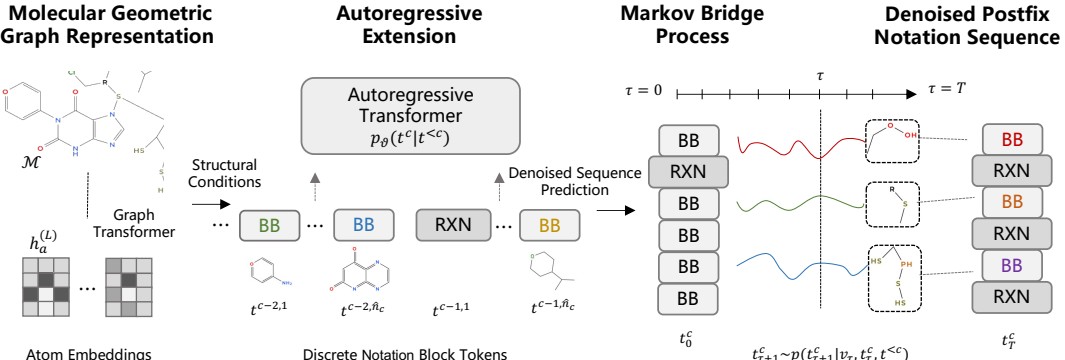

**Figure 1:** Overview of MoleBridge. MoleBridge sequentially extends postfix notation tokens from molecular graphs conditioning on previous blocks. By performing Markov bridge, MoleBridge retains scalable capability for the increasing synthetic space and supports higher-quality generation.

are composed of a limited number of building blocks, with minimal redundancy and a lack of semantic coherence. Therefore, the model should pay attention to the deep-level connections between postfix notations. iii) Third, early errors in pathway generation propagate irreversibly. In multi-step synthesis, selecting an incompatible building block at a certain step constrains all subsequent reactions, potentially invalidating the entire pathway. This necessitates a generation mechanism that allows progressive refinement of pathway segments.

In this paper, we introduce MoleBridge, a novel Markov bridge generative model for designing postfix notations of molecular synthesis pathways. Markov bridge models exhibit increased flexibility (the noise addition and removal processes resemble powerful data augmentation, compelling the model to form deeper relationships among features). However, constrained by the noise scheduling process, they follow a fixed generation length [3, 73, 52], making it difficult to explore synthetic pathways of variable lengths. Thus, instead of adding Gaussian noise to an entire fixed-length synthesis pathway, MoleBridge introduces probabilistic perturbations to notation token blocks at current autoregressive steps. By approximately referencing the Markov bridge process [17, 29, 78], MoleBridge learns to progressively refine perturbed reaction sequences, thereby generating more rational designs. In pursuit of stable pathways and semi-autoregressive error robustness, we identify errors introduced in the sequence steps by thinking at each denoising time step and correct these errors based on the selected positions in the current noisy sequence block. This innovation is more natural for molecular synthesis pathways than for language data. Empirically, we demonstrate that MoleBridge excels in a variety of scenarios.

Our contributions can be summarized as follows:

- We introduce MoleBridge, a novel generative approach for generating molecular postfix notations based on the Markov bridge, which expands chemical space via a semi-autoregressive process while applying iterative refinement to each block sequentially.

- We employ a thinking strategy, which performs denoising when an error occurs at a certain step, ensuring the feasibility of the synthesis pathway.

- Through experiments, we demonstrate the effectiveness of MoleBridge in various scenarios, such as bottom-up synthesis, structure-based drug design, and target-directed generation.

## 2   Related Work

**Synthesizability of molecules.** Synthesizable molecule design aims to generate novel molecular structures that can be realized through practical chemical synthesis pathways. Early approaches [60, 24] relied on proposing a large number of potential candidate molecules and screening them using scoring functions to estimate energy and identify stable molecules. Groundbreaking deep

learning approaches [6, 19, 7] have now been developed to predict reaction outcomes in a "template-free" manner. For instance, MoleculeChef [6] encodes and decodes a set of initial reactants from purchasable building blocks, enumerates possible one-step synthetic paths, and selects the best molecule among the products as the output. It allows chemists to interrogate the properties of the generated molecules. DoGs [7] employs a recurrent neural network to generate sequences of actions from latent codes, achieving molecular generation through sampling in the latent space. It also demonstrates strong capabilities to satisfy the feasibility of the synthesis paths. However, empirical validation remains challenging, since they are generally less effective at producing convergent synthesis paths and structurally complex molecules. The synthetic space projecting [46] aims to generate structurally similar and synthesizable analogs by converting them into synthetic postfix notations. It is capable of guaranteeing bottom-up synthesis planning and exploring the locally synthesizable chemical space around hit molecules. Despite this progress, directly applying these approaches to postfix-notation path generation remains challenging and requires further design.

**Diffusion generative models.** Diffusion models [12, 64] are generative models achieving state-of-the-art performance across various domains, including the generation of images [62, 44], video [66], or molecular [70, 69]. It achieves high-quality and diverse sampling from unknown data distributions by approximating the simple density (i.e., Gaussian density) to the stochastic differential equation of the unknown data density [25]. A notable highlight of the success of diffusion models in the field of molecular design is their potential to generate molecules that can serve as the foundation for novel medicinal compounds previously unseen [32]. Multiple approaches have been explored to achieve this. For instance, 3D diffusion methods such as MDM [28], DiffLinker [30], and PIDiff [13] can generate candidate drug molecules relevant to chemistry. Most relevant to our work, diffusion models have also achieved promising results in designing discrete DNA sequences [61] and optimizing stable molecules [35, 76, 55]. Recently, some studies [43, 45] propelled the development of molecular design technologies by utilizing more reliable diffusion processes.

**Schrödinger bridge.** The Schrödinger Bridge (SB) problem [53, 4, 38] arises from an intriguing connection between statistical physics and probability theory. Its goal is to find the most probable evolution between a given initial and final distribution relative to a specified reference stochastic process [72]. A significant characteristic of the SB problem is the ability to choose any distribution as the initial and terminal distributions [33], which has advanced the resolution of various generative model issues. Building upon the SB framework, diffusion bridge models have shown cutting-edge results in various fields, including imaging [15, 36], speech [37], and physical fields [41]. The recently proposed Markov bridge models [78, 29, 17] extend these models into the discrete domain, focusing on environments with categorical distributions. In this work, we apply the Markov bridge for molecular postfix notation path synthesis.

## 3 Preliminary

### 3.1 Notations and Problem Formulation

The synthetic space is constructed by recursively applying reaction rules to all possible molecular combinations, starting from the initial building blocks [46]. Mathematically, a synthetic space $\mathcal{S}$ is represented as the closure of molecules generated by a set of $n_b$ building blocks $\mathcal{B} = \{\boldsymbol{b}^1, \boldsymbol{b}^2, \ldots, \boldsymbol{b}^{n_b}\} \in \mathcal{S}$ and a set of $n_r$ reaction rules $\mathcal{R} = \{\boldsymbol{r}^1, \boldsymbol{r}^2, \ldots, \boldsymbol{r}^{n_r}\}$, where each reaction rule $\boldsymbol{r}^i$ defines a mapping function from the reactant space to the product: $\boldsymbol{r}^i := \mathcal{X} \times \mathcal{Y} \to \mathcal{S}, (\mathcal{X}, \mathcal{Y}) \mapsto \mathcal{Z}$. Here, $\mathcal{X}, \mathcal{Y} \in \mathcal{S}$ represents the sets of molecules applicable to the reaction $\boldsymbol{r}^i$, and $\mathcal{Z}$ represents the main reaction product. In the synthetic space construction process, synthesis pathways are represented as notation sequences $\boldsymbol{y} = [\boldsymbol{t}^1, \boldsymbol{t}^2, \ldots, \boldsymbol{t}^{n_t}]$, where each token $\boldsymbol{t}^\ell \in \mathcal{B} \cup \mathcal{R}$ indicates pushing a building block onto the stack or calculating the product and pushing it back onto the stack, with $n_t$ indicating the length of the tokens. The goal of the synthetic space projecting problem is to identify a structurally similar and practically synthesizable analog $\boldsymbol{y}$ by projecting the designed molecule $\mathcal{M}$ into the synthesizable chemical space $\mathcal{S}$. Thus, an ideal model, parameterized by $\vartheta$, should be capable of learning the mapping from any molecule $\mathcal{M}$ to its corresponding postfix notation distribution $p_\vartheta(\boldsymbol{y} \mid \mathcal{M})$.

### 3.2 Autoregressive Models

The success of large models in the natural language processing and computer vision domain demonstrates their scalability and the universality of sequence data modeling [3, 68, 65, 75]. Given a

postfix notation sequence $\boldsymbol{y} = [\boldsymbol{t}^1, \boldsymbol{t}^2, \ldots, \boldsymbol{t}^{n_t}]$, where the subscript $1 \leq \ell \leq n_t$ specifies an order, autoregressive models assume that the probability of observing the current token $\boldsymbol{t}^\ell$ depends only on its prefix $[\boldsymbol{t}^1, \boldsymbol{t}^2, \ldots, \boldsymbol{t}^{\ell-1}]$. This unidirectional token dependency assumption allows the likelihood of the sequence $\boldsymbol{y}$ to be factorized as:

$$\log p_\vartheta(\boldsymbol{y}) = \sum_{\ell=1}^{n_t} \log p_\vartheta(\boldsymbol{t}^\ell \mid \boldsymbol{t}^{<\ell}), \tag{1}$$

where $p_\vartheta(\boldsymbol{t}^\ell \mid \boldsymbol{t}^{<\ell})$ is parameterized directly with a neural network. Therefore, autoregressive models can be efficiently trained through "next token prediction" [54, 71, 74]. However, due to sequential dependencies, autoregressive models require $n_t$ steps to generate $n_t$ tokens.

## 3.3 Discrete Markov Bridge

Markov bridge model fits a model $\psi_\vartheta(\cdot)$ to reverse the forward corruption process $q$ [78, 29, 17]. This process is pinned to specific data points in the beginning and in the end, modeling the dependencies between the discrete spaces $X$ and $Y$. For the sample pair $(\boldsymbol{x}, \boldsymbol{y}) \sim p_{X,Y}(\boldsymbol{x}, \boldsymbol{y})$ and the time step sequence $\tau = 0, 1, \ldots, T$, it defines the Markov process as a sequence of random variables $(\boldsymbol{t}_\tau)_{\tau=0}^T$, starting from $\boldsymbol{t}_0 = \boldsymbol{x}$, and satisfying the Markov property:

$$p(\boldsymbol{t}_\tau \mid \boldsymbol{t}_0, \boldsymbol{t}_1, \ldots, \boldsymbol{t}_{\tau-1}, \boldsymbol{y}) = p(\boldsymbol{t}_\tau \mid \boldsymbol{t}_{\tau-1}, \boldsymbol{y}). \tag{2}$$

To ensure the process terminates at the data point $\boldsymbol{t}_T = \boldsymbol{y}$, we introduce an additional requirement:

$$p(\boldsymbol{t}_T = \boldsymbol{y} \mid \boldsymbol{t}_{T-1}, \boldsymbol{y}) = 1. \tag{3}$$

Suppose the distributions $p_X$ and $p_Y$ are categorical distributions with a finite sample space $1, \ldots, K$, and we can represent the data points as $K$-dimensional one-hot vectors: $\boldsymbol{x}, \boldsymbol{y}, \boldsymbol{t}_\tau \in \mathbb{R}^K$, and define the transition probabilities (Eq. (2)) as follows:

$$p(\boldsymbol{t}_{\tau+1} \mid \boldsymbol{t}_\tau, \boldsymbol{y}) = \mathrm{Cat}(\boldsymbol{t}_{t+1}; \boldsymbol{Q}_\tau \boldsymbol{t}_\tau), \tag{4}$$

where $\mathrm{Cat}(\cdot; \boldsymbol{p})$ is a categorical distribution with probabilities given by $\boldsymbol{p}$, and $\boldsymbol{Q}_\tau$ is a transition matrix parameterized as:

$$\boldsymbol{Q}_\tau := \boldsymbol{Q}_\tau(\boldsymbol{y}) = \alpha_\tau \boldsymbol{I}_K + (1 - \alpha_\tau)\boldsymbol{y}\boldsymbol{1}_K^\top, \tag{5}$$

where $\boldsymbol{I}_K$ denotes a $K \times K$ identity matrix, and $\alpha_\tau$ is a schedule parameter transitioning from $\alpha_0 = 1$ to $\alpha_{T-1} = 0$. Then, $\boldsymbol{t}_\tau$ can be efficiently sampled from $p(\boldsymbol{t}_{\tau+1} \mid \boldsymbol{t}_0, \boldsymbol{t}_T) = \mathrm{Cat}(\boldsymbol{t}_{\tau+1}; \bar{\boldsymbol{Q}}_\tau \boldsymbol{t}_0)$, where $\bar{\boldsymbol{Q}}_\tau = \boldsymbol{Q}_\tau \boldsymbol{Q}_{\tau-1} \ldots \boldsymbol{Q}_0 = \bar{\alpha}_\tau \boldsymbol{I}_K + (1 - \bar{\alpha}_\tau)\boldsymbol{y}\boldsymbol{1}_K^\top$ a cumulative product matrix, and $\bar{\alpha}_\tau = \prod_{s=0}^\tau \alpha_s$. During training, the Markov bridge approximates $\boldsymbol{y}$ using the neural network $\psi_\vartheta$: $\hat{\boldsymbol{y}} = \psi_\vartheta(\boldsymbol{t}_\tau, \tau)$.

# 4 Methods

In this section, we introduce MoleBridge, a Markov bridge model for synthetic space projecting. We first define how to interpolate between semi-autoregressive and Markov bridge models by defining autoregressive distributions over tokens. Next, we provide an objective for maximum likelihood estimation and efficient training and sampling algorithms. Finally, we describe the denoising network architecture employed to approximate the Markov bridges.

## 4.1 Markov Bridge for Synthetic Space Projecting

Generally, molecular synthesis paths are highly discrete processes that strictly adhere to chemical rules. A common scenario is to construct the pathway from simple building blocks to complex target molecules through sequential decisions alone. In this work, we propose to introduce Markov bridges into molecular synthesis pathways, modeling blocks of tokens autoregressively. We group the postfix notation tokens into $\mathcal{C}$ notation blocks, each of length $\hat{n}_c$, where $\mathcal{C} = n_t/\hat{n}_c$ (assuming $\mathcal{C}$ is an integer). We denote each block $\boldsymbol{t}_{:c\hat{n}_c}$ from token at positions 0 to $c\hat{n}_c$ for blocks $c \in \{1, \ldots, \mathcal{C}\}$ as $\boldsymbol{t}^c$ for simplicity.

**Interpolation process.** As discussed in Section 3.3, the matrix $\boldsymbol{Q}_\tau$ can efficiently model various transition probabilities in the discrete state space, including masking, random token changes, and

related word substitutions. When considering the noise process modulated by the masking vector [78, 29], the interpolation process gradually transforms the data point $t_T^{1:}$ into the initial molecular state $t_0^{1:} \sim p_0(t)$, with the transition rate controlled by the masking vector. In this case, the noise interpolation can be represented as:

$$q(t_\tau^{1:} \mid v_\tau, t_0^{1:}, y) = v_\tau t_0^{1:} + (1 - v_\tau)y, \tag{6}$$

where $v_\tau \sim \text{Bernoulli}(\bar{\beta}_{\tau-1})$ is a masking latent vector and $\beta_\tau$ denotes a scheduler. At $\tau = 0$, the conditional marginal converges to the initial distribution $t_0^{1:}$, i.e. $\beta_\tau = 1$. When $\tau \to T$, $\beta_\tau$ is set close to 0 and the distribution is closer to the target distribution.

**Reverse process.** Accordingly, the reverse process in block $c$ can be written as:

$$p_\vartheta(t_{\tau+1}^c \mid v_\tau, t_\tau^c, t^{<c}) = \sum_{t^c} q(t_{\tau+1}^c \mid v_\tau, t_\tau^c, t^c) p_\vartheta(t^c \mid v_\tau, t_\tau^c, t^{<c}), \tag{7}$$

where the denoising base model predicts clean token $t^c$ given the noisy sequence $t_\tau^c$. Since masked diffusion requires building a complete probability distribution $p(t^c \mid t_\tau^c)$ over all possible values at each position $\ell \in \{1, \ldots, |t^c|\}$ rather than directly identifying corrupted locations $\{\ell \mid t_\tau^{c,\ell} \neq t^{c,\ell}\}$, it is difficult to achieve accurate denoising probability estimation [56, 47]. As evidenced by recent works [11, 56, 51], the core difficulty stems from denoising probability estimation $p_\vartheta$ being the only key component in the diffusion process that requires neural network approximation. Unfortunately, while uniform diffusion [45] allows token values to be corrected throughout the sampling process, by modeling the posterior $q(t_{\tau+1}^c \mid v_\tau, t_\tau^c, t^c)$ as shown in Eq. (7), its performance does not consistently outperform masked diffusion, particularly in tasks like image or language modeling.

Indeed, the transition probability can be decomposed into scheduling probability, which assesses whether the data has been corrupted, and a denoising probability that determines the new value [45]. Therefore, we first think the probability of each position in the sequence being corrupted by noise [45], conditioned on the current state $t_\tau^c$ and time step $\tau$, and define:

$$p_\vartheta(t^c \mid v_\tau, t_\tau^c, t^{<c}) = \begin{cases} t_\tau^c, & \text{if } v_\tau = 0 \\ \frac{\dot{\beta}_\tau}{1-\beta_\tau} \cdot p_\theta\left(z_\tau^c = 1 \mid t_\tau^c\right) \cdot \left((1 - \beta_\tau)\psi_\vartheta(t_\tau^c, \tau) + \beta_\tau t_\tau^c\right), & \text{if } v_\tau = 1 \end{cases} \tag{8}$$

where $p_\theta(z_\tau^c = 1 \mid t_\tau^c)$ denotes the probability that each position is corrupted, and $z_\tau^c \in \{0,1\}^{\hat{n}_c}$ is a latent variable to denote if a dimension is corrupted. If a position is likely to be corrupted ($p_\theta(z_\tau^c = 1 \mid t_\tau^c) \approx 1$), the transition probability will lean toward the output of the denoiser. On the other hand, when a position is likely to be clean ($p_\theta(z_\tau^c = 1 \mid t_\tau^c) \approx 0$), the transition probability will tend to preserve the current state. In mask diffusion case, this can be readily read out from the masked token. But in the uniform diffusion case, we need to compute/approximate this probability instead.

**Training.** We obtain a principled learning objective for the model $\psi_\vartheta(\cdot)$ by applying the variational bound on negative log-likelihood $\log p_\vartheta(y \mid t)$ to each term, which has the following closed-form expression:

$$-\log p_\vartheta(t) \leq \mathcal{L}_\tau(t, \vartheta) := \sum_{c=1}^{\mathcal{C}} \mathbb{E}_{q(t_\tau^c \mid t^c, t^{<c}, y^c)}\left[-v_\tau y^{cT} \log \psi_\vartheta(t_\tau^c, \tau)\right], \tag{9}$$

where $y^c$ denotes the true tokens in block $c$. The derivation of $\mathcal{L}_\tau(t, \vartheta)$ expresses the training loss as a reweighted standard multiclass cross-entropy loss [78, 26], which is computed on the labels that have not yet been converted into the base truth $y = t_T$. Compared to the simpler cross-entropy loss computed over all labels, this new approach assigns greater weight to the labels that need refinement. Since the model is conditioned on $t^{<c}$, the dependency between $t^{<c}$ and $\vartheta$ is explicated in $\mathcal{L}_\tau$.

The training of $p_\theta$ can be simplified to a binary classification task, aiming to estimate the probability that each position is corrupted by noise. Specifically, the training objective is formulated as:

$$\mathcal{L}_p(t, \theta) = \sum_{c=1}^{\mathcal{C}} \mathbb{E}_{\tau \sim \mathcal{U}(0,T)}\left[-\frac{\dot{\beta}_t}{1-\beta_t}\sum_{\ell=1}^{\hat{n}_c} \text{BCE}\left(p_\theta(z_\tau^{c,\ell} = 1 \mid t_\tau^{c,\ell}), \mathbb{I}(t_\tau^{c,\ell} \neq y^{c,\ell})\right)\right], \tag{10}$$

where $z_\tau^{c,\ell}$ is a variable indicating whether the $\ell$-th position in block $c$ is corrupted at time step $\tau$, $t_\tau^{c,\ell}$ is the noisy token at that position, $y^{c,\ell}$ is the true value, $\mathbb{I}(\cdot)$ is the indicator function, and $\text{BCE}(\cdot)$ represents binary cross-entropy. The final loss function is the sum of the two terms: $\mathcal{L} = \mathcal{L}_\tau + \mathcal{L}_p$.

**Sampling.** A postfix notation of synthesis $\boldsymbol{y}$ is a sequence that contains four types of tokens: building block tokens $\boldsymbol{b}^j \in \mathcal{B}$, reaction tokens $\boldsymbol{r}^i \in \mathcal{R}$, a start token [START], and an end token [END]. Each building block token $\boldsymbol{b}^j$ is associated with the fingerprint of the corresponding molecule, represented as [BB,j], where j$\in \{0,1\}^{256}$ is the Morgan fingerprint of length 256 and radius 2 [46, 49]. A reaction token $\boldsymbol{r}^i$, denoted as [RXN,i], represents the index i of the reaction. During sampling, we generate the pathway token-by-token, and each token is produced through a block-wise Markov bridge refinement process. The conditional distribution $p_\vartheta(\boldsymbol{t}_{\tau+1}^c \mid \boldsymbol{v}_\tau, \boldsymbol{t}_\tau^c, \boldsymbol{t}^{<c})$ is used to sample from the model. Starting from the given $\boldsymbol{t}_0^c \sim p_0(\boldsymbol{t}^c)$, the process iterates to predict the data point $\hat{\boldsymbol{y}}^c = \psi_\vartheta(\boldsymbol{t}_\tau^c, \tau)$ and then derive $\boldsymbol{t}_{\tau+1}^c \sim p_\vartheta(\boldsymbol{t}_{\tau+1}^c \mid \boldsymbol{v}_\tau, \boldsymbol{t}_\tau^c, \boldsymbol{t}^{<c})$ while incrementing the time step $\tau$ from 0 to $T-1$. Notably, our algorithm allows us to sample sequences of arbitrary lengths, whereas traditional Markov bridge models are limited to fixed-length generation.

In the synthesis process, the postfix stack is initialized as empty and is progressively populated by the generated tokens [46]. When a building block token $\boldsymbol{b}^j$ is generated, we retrieve the corresponding molecule from the fingerprint and push it onto the stack. If a reaction token $\boldsymbol{r}^i$ is generated, we first pop the required number of molecules from the stack, then use the reaction template with RDKit [8] to predict the product, which is subsequently pushed onto the stack. If there are insufficient molecules on the stack or if the reaction cannot be applied, the inference process halts. Finally, the process ends when the [END] token is generated, marking the completion of the synthesis. The most recent product molecule is then considered as the input molecule for the synthetic space projecting.

## 4.2 Architecture Design

**Molecular graph representation.** Following [46], we represent a molecule $\mathcal{M}$ as a graph, where nodes are connected based on chemical bonds. For atoms, we convert them into initial embedding vectors $\boldsymbol{h}_a^{(0)} \in \mathbb{R}^d$ based on their atomic numbers; for chemical bonds, we capture bond type information and incorporate it into the graph structure, represent as $\boldsymbol{h}_e^{(0)} \in \mathbb{R}^{de}$.

**Networks.** We use a transformer [58] as the backbone network to approximate the final state of the Markov bridge process. Typically, a transformer only takes the synthesis path sequence as input, but our task also requires integrating time steps and structural conditions into the model. The time-step embedding network encodes temporal information using sinusoidal functions, converts it into vector $\boldsymbol{e}_\tau \in \mathbb{R}^d$, and integrates it into the cross-attention mechanisms. We use a $L$-layer graph transformer [67] to capture the molecular topology and inter-atomic interactions, with the final atomic representations $\boldsymbol{h}_a^{(L)}$ serving as structural information. Finally, using the customized transformer decoder architecture, the network takes both sequence embeddings and molecular graph embeddings as input. For the thinking network $p_\theta(\boldsymbol{z}_\tau^c = 1 \mid \boldsymbol{t}_\tau^c)$, we use a multilayer perceptron with $\mathrm{gelu}(\cdot)$ activation and a $\mathrm{sigmoid}(\cdot)$ output layer. The entire architecture is trained end-to-end, and the output types include BB, RXN, and END, which together form the complete molecular synthesis pathway.

# 5 Experiments

## 5.1 Experimental Setup

**Datasets.** We use the SynNet reaction template set [19] for reaction templates $\mathcal{R}$, which is based on two publicly available template collections from Hartenfeller et al [24]. and Button et al [10]. After removing duplicates and rare reactions, a final set of 91 reaction templates is obtained. The set includes 13 unimolecular and 78 bimolecular reactions. For building blocks $\mathcal{B}$, we use the Enamine US Stock catalog [1] as the data source. Entries containing multiple molecules (e.g., salts or hydrates) are filtered by retaining the largest molecule and removing the rest. Any building blocks that fail RDKit sanitization or do not match any reaction templates are excluded, as are duplicates. We use $K$-means clustering based on Morgan fingerprints to group the blocks into 128 clusters, reserving one structurally distinctive cluster for testing and using the remaining 127 clusters for training. Additionally, we include a challenging test set: molecules extracted from the ChEMBL database [20], which have been previously reported as "unreachable" target compounds [19, 46].

**Implementation details.** In our experiments, we use Morgan fingerprints [49] to featurize molecular structures, with a radius of 2 and a bit length of 256. At the data initialization stage, we randomly initialize the reaction path stack using weighted sampling, assigning an initial weight of 0.90 to

**Table 1:** Performance comparison between MoleBridge and baseline methods. The evaluation is performed on both the standard test set and the ChEMBL [20]. Best results are highlighted in **bold**.

| Dataset | Method | Success (↑) | Recons. %(↑) | Sim.(Morgan) (↑) | Sim.(Scaffold) (↑) | Sim.(Gobbi) (↑) |
|---------|--------|-------------|--------------|------------------|--------------------|-----------------|
| Test Set | SynNet | 0.4205 | 10.7% | 0.4575 | 0.5109 | 0.3465 |
|  | ChemProjector | 0.4875 | 28.4% | 0.7167 | 0.7791 | 0.7273 |
|  | **MoleBridge (Ours)** | **0.4915** | **43.5%** | **0.8455** | **0.8695** | **0.8287** |
| ChEMBL | SynNet | 0.4250 | 5.4% | 0.4270 | 0.4174 | 0.2678 |
|  | ChemProjector | 0.4940 | 13.3% | 0.5978 | 0.5869 | 0.5570 |
|  | **MoleBridge (Ours)** | **0.4970** | **14.6%** | **0.6159** | **0.6188** | **0.5789** |

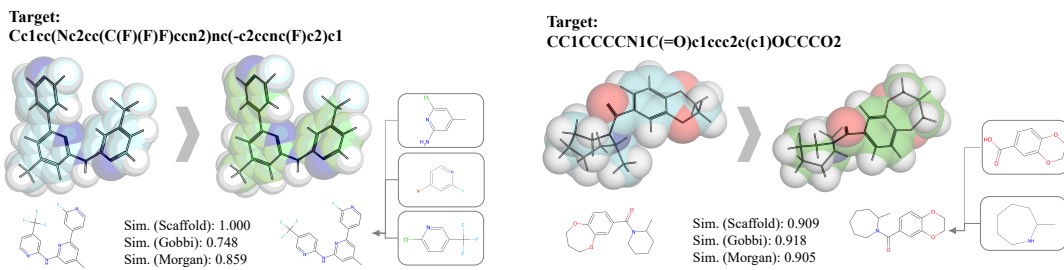

**Target:**
**Cc1cc(Nc2cc(C(F)(F)F)ccn2)nc(-c2ccnc(F)c2)c1**

Sim. (Scaffold): 1.000
Sim. (Gobbi): 0.748
Sim. (Morgan): 0.859

**Target:**
**CC1CCCCN1C(=O)c1ccc2c(c1)OCCCO2**

Sim. (Scaffold): 0.909
Sim. (Gobbi): 0.918
Sim. (Morgan): 0.905

**Figure 2:** Examples of molecules generated by Pocket2Mol and projected by MoleBridge into analogs, demonstrating clear synthetic pathways and the preservation of high structural similarity.

building blocks. The model is trained on $4$ NVIDIA $4090$ GPUs with a batch size of $128$ and $4$ data loader workers. We use the Adam optimizer [34] with an initial learning rate of $3 \times 10^{-4}$, and momentum parameters $\beta_1 = 0.90$, $\beta_2 = 0.999$. A plateau-based learning rate scheduler is used, reducing the learning rate by a factor of $0.6$ when validation performance plateaus, with patience of $5$ validation cycles and a minimum learning rate of $1 \times 10^{-5}$.

**Metrics.** We conduct a comprehensive evaluation of the MoleBridge model using multiple quantitative metrics: i) Synthesis path success rate: the percentage of valid postfix notations and further multiplied by 1/2. ii) Reconstruction rate: the percentage of proposed synthesis paths that result in the same product as the input molecule. iii) In cases of partial success, we evaluate the molecular similarity between the generated and target compounds; a similarity score of $0$ is assigned to failed (invalid) syntheses. The similarity score is calculated using three types of fingerprint representations: Morgan fingerprints of length $4096$ and radius $2$ [49], Murcko scaffold-based fingerprints, and Gobbi pharmacophore fingerprints [22]. All three similarity scores are normalized to the $[0, 1]$, reflecting chemical similarity in terms of overall structure, scaffold structure, and pharmacophoric properties.

## 5.2 Bottom-Up Synthesis Planning

To validate the effectiveness of MoleBridge, we conduct a comparison with the existing generation methods, SynNet [19] and ChemProjector [46]. The results presented in Table 1 show that MoleBridge outperforms the baseline methods significantly on all evaluation metrics. A particularly notable result is the reconstruction rate on the test set, where MoleBridge achieved $43.5\%$, significantly outperforming ChemProjector ($28.4\%$) and SynNet ($10.7\%$). Even on the challenging ChEMBL dataset, MoleBridge retains its superior performance.

## 5.3 Projecting Molecules Generated by Structure-Based Drug Design Models

Synthetic space projecting has broad application prospects in the field of structure-based *de novo* drug design. Due to limited constraints [23], existing design models often generate chemically invalid structures [46]. To assess the applicability of MoleBridge in drug optimization scenarios, we conduct experiments based on the LIT-PCBA dataset [57], which contains $15$ drug targets. Following [46],

**Table 2:** Similarity scores between molecules generated by Pocket2Mol and their analogs.

| Targets | Sim. (Morgan) | Sim. (Scaffold) | Sim. (Gobbi) |
|---|---|---|---|
| ADRB2 | 0.5149 | 0.5834 | 0.4007 |
| ALDH1 | 0.4434 | 0.3535 | 0.3304 |
| ESR1 ago | 0.4641 | 0.3690 | 0.2993 |
| ESR1 ant | 0.5078 | 0.4903 | 0.4229 |
| FEN1 | 0.4397 | 0.4308 | 0.3408 |
| GBA | 0.4267 | 0.2785 | 0.2572 |
| IDH1 | 0.4701 | 0.3642 | 0.3224 |
| KAT2A | 0.5123 | 0.4927 | 0.4545 |
| MAPK1 | 0.4990 | 0.3955 | 0.3917 |
| MTORC1 | 0.5351 | 0.4384 | 0.3841 |
| OPRK1 | 0.5170 | 0.5500 | 0.4560 |
| PKM2 | 0.4874 | 0.4521 | 0.3927 |
| PPARG | 0.4977 | 0.4904 | 0.4616 |
| TP53 | 0.5289 | 0.5595 | 0.4979 |
| VDR | 0.5312 | 0.3990 | 0.3730 |

**Table 3:** The analogs optimized by model exhibit a significant increase in Vina scores.

| Targets | Vina (kcal/mol) | | | |
|---|---|---|---|---|
| | Ref. | Gen. | Analog ($\downarrow$) | $\Delta$ |
| ADRB2 | -8.70 | -8.31 | -10.90 | -2.59 |
| ALDH1 | -5.20 | -8.14 | -11.00 | -2.86 |
| ESR1 ago | -5.90 | -8.23 | -9.30 | -1.07 |
| ESR1 ant | -8.10 | -8.77 | -11.80 | -3.03 |
| FEN1 | -5.80 | -6.04 | -6.60 | -0.56 |
| GBA | -8.40 | -6.86 | -6.96 | -0.10 |
| IDH1 | -9.30 | -8.62 | -9.70 | -1.08 |
| KAT2A | -8.00 | -7.41 | -11.40 | -3.99 |
| MAPK1 | -8.80 | -8.20 | -9.90 | -1.70 |
| MTORC1 | -8.80 | -9.32 | -12.10 | -2.78 |
| OPRK1 | -9.00 | -7.88 | -10.30 | -2.42 |
| PKM2 | -9.20 | -8.25 | -11.60 | -3.35 |
| PPARG | -7.70 | -7.60 | -9.10 | -1.50 |
| TP53 | -6.30 | -6.83 | -9.80 | -2.97 |
| VDR | -8.40 | -9.27 | -10.30 | -1.03 |

we use Pocket2Mol [50] to generate candidate molecules for each target and select the top 300 candidates for each target based on QED and SA scores. Subsequently, MoleBridge is applied to design 5 analogs for each candidate molecule, and the optimal analogs are selected based on Vina scores [16]. Figure 2 shows examples of molecular synthesis pathways generated by MoleBridge, illustrating the complete synthesis pathway design from Pocket2Mol to the target analog. Table 2 presents the structural similarity metrics between the generated analogs and the original molecules. Table 3 displays the estimated binding energies, with optimized analogs demonstrating improved target binding strength.

## 5.4 Projecting Molecules to Explore Local Chemical Space for Hit Expansion

MoleBridge also demonstrates significant application potential in hit expansion. Following the experimental design of Levin et al. [39] and Luo et al. [46], we evaluate the development of c-Jun N-terminal Kinases-3 (JNK3) inhibitors. Using an molecule with a JNK3 score [40] of 0.68 as the starting point, MoleBridge successfully generates several synthesizable structural analogs as in Figure 3. Notably, the generated analogs effectively preserve the JNK3 inhibitory activity (0.67, 0.68, and 0.70) while maintaining the integrity of the core scaffold structure. Taking the amino-substituted analog in the lower-left corner as an example, the introduction of an amino functional group not only maintained almost the

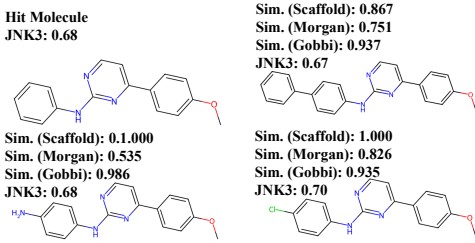

**Figure 3:** Analogs generated from the initial hit expansion compound.

same JNK3 activity (0.68) but also potentially provided new modification sites for further structural optimization.

## 5.5 Projecting Molecules Generated by Target-Directed Generative Models

Target-directed generative models generally suffer from insufficient synthesize-ability, with approximately 70% of generated molecules being labeled as non-synthesizable by the ASK-COS [14, 18].

**Table 4:** Assessment of property retention after projection of molecules generated by target-directed.

| Property | Sim. Morgan | Sim. Scaffold | Sim. Gobbi | Avg(Objective) Gen. | Analog | Δ | Max(Objective) Gen. | Analog | Δ |
|---|---|---|---|---|---|---|---|---|---|
| Amlodipine MPO | 0.55224 | 0.3734 | 0.4807 | 0.84384 | 0.7600 | -0.0838 | 0.8894 | 0.8793 | -0.0101 |
| Deco Hop | 0.52903 | 0.8650 | 0.8180 | 0.9712 | 0.8633 | -0.1078 | 0.9992 | 0.9651 | -0.0340 |
| Fexofenadine MPO | 0.45537 | 0.487 | 0.4736 | 0.94273 | 0.7813 | -0.1613 | 1.0000 | 0.8545 | -0.1454 |
| Osimertinib MPO | 0.43848 | 0.4830 | 0.6339 | 0.9148 | 0.8484 | -0.0663 | 0.9508 | 0.9140 | -0.0367 |
| Perindopril MPO | 0.43919 | 0.4344 | 0.5119 | 0.6832 | 0.6413 | -0.0419 | 0.7733 | 0.7374 | -0.0358 |
| Ranolazine MPO | 0.34952 | 0.4362 | 0.4460 | 0.8868 | 0.4506 | -0.4361 | 0.9102 | 0.8310 | -0.0792 |
| Scaffold Hop | 0.3958 | 0.5879 | 0.5646 | 0.9467 | 0.5401 | -0.4065 | 1.0000 | 0.8345 | -0.1654 |
| Sitagliptin MPO | 0.28301 | 0.2629 | 0.2825 | 0.5747 | 0.0848 | -0.4898 | 0.8315 | 0.4909 | -0.3406 |
| Valsartan SMARTS | 0.34513 | 0.3568 | 0.3060 | 0.8331 | 0.0452 | -0.7878 | 0.9860 | 0.9283 | -0.0577 |
| Zaleplon MPO | 0.522717 | 0.7238 | 0.62105 | 0.6384 | 0.4668 | -0.1715 | 0.7150 | 0.7071 | -0.0079 |

Next, we evaluate the application potential of MoleBridge in this scenario. We generate synthesizable alternative structures for these molecules that were deemed non-synthesizable. As shown in Table 4, the experimental results reveal that MoleBridge not only guarantees the synthesizability of generated molecules but also strikes a balance in property degradation. For instance, in the Amlodipine MPO task, the average objective function value decreased by only $0.0838$, while the optimal molecular property remained at $0.8793$, nearly the same as the original value.

## 5.6 Ablation Studies

We conduct ablation experiments on the test set to validate the key contributions of thinking and cross-attention in MoleBridge's performance, as shown in Table 5. The cross-attention ensures the effective transfer of information across different modalities, while the thinking mechanism optimizes the execution of the denoising process. Together, they collaborate to achieve optimal generation performance.

**Table 5:** Ablation results of components. The cross-attention mechanism is replaced by fully connected layers when disabled.

| Mechanism w/ thinking | Network w/ cross-attention | All Success | Recons. | Sim Morgan | Scaffold | Gobbi |
|---|---|---|---|---|---|---|
| ✗ | ✓ | 0.4840 | 40.1% | 0.8266 | 0.8470 | 0.8068 |
| ✓ | ✗ | 0.4820 | 41.1% | 0.8367 | 0.8338 | 0.8168 |
| ✓ | ✓ | **0.4915** | **43.5%** | **0.8455** | **0.8695** | **0.8287** |

## 6 Conclusion and Limitations

In this paper, we introduce MoleBridge, a novel semi-autoregressive Markov bridge process for synthetic space projecting. Experimental results demonstrate that MoleBridge excels in various scenarios, such as bottom-up synthesis, structure-based drug design, target-directed generation, and hit expansion. Despite significant progress, MoleBridge still has **some limitations**. First, the optimization space is limited for molecular structures that contradict existing synthesis logic. Second, for some synthetic paths with specific stereoselective requirements, the model's control ability needs improvement. **Future work** will expand the reaction template library to cover more types of chemical transformations.

## 7 Broader impacts

Synthetic space projecting technology has a profound impact on the field of *de novo* molecular design by converting molecules that lack synthetic pathways into structurally similar and synthesizable analogs. This approach can significantly accelerate the process from virtual screening to clinical candidates, reducing the high failure rate in traditional drug development due to synthesis barriers.

# 8 Acknowledgments

This paper was supported by National Key R&D Program of China (No. 2023YFC3502902, 2021YFF1201100), National Natural Science Foundation of China under Grants (62436006), Sanya Science and Technology Special Fund (No. 2024KFJX04), Beijing Natural Science Foundation (No. L257018) and Beijing Nova Program.

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
