# OpenReview forum: "MoleBridge: Synthetic Space Projecting with Discrete Markov Bridges"
_NeurIPS.cc/2025/Conference — NeurIPS 2025 poster_

### Official Review · Reviewer_bSFf · 2025-06-30

**Clarity:** 4
**Significance:** 3
**Originality:** 3
**Rating:** 5
**Confidence:** 4

**Summary:**

This paper proposes MoleBridge, a technique for predicting a sequence of postfix notations for synthesizing molecules from a database of available building blocks and reactions in the context of chemical space projection. To enable inference of variable length sequences, the authors propose modelling groups of tokens autoregressively, where each group of tokens undergoes a discrete denoising diffusion process by first predicting which position is to be denoised at each iteration, and then denoising that position. The performance of the resulting model is good, beating existing state-of-the-art on a test set (obtained from Enamine US Stock catalog after clustering) and ChEMBL.

**Questions:**

- I understand the grouping of tokens (equation 6) as necessary from an efficiency/variable length prediction perspective. How does varying the group size affect the performance of your model? My understanding is that with group size = 1, you should recover the fully autoregressive case. Do you have results for that?
- Line 178: the notation ($t^{1:}_T$) is a bit unclear (the “${1:}$” part). Could you explain that better in the text? Is it because you start from a given molecule which corresponds to $t^{0}$?
- Equation 7: second $v_\tau$ should be bold (I think)
- Equation 9: should $\beta_t $ be $\beta_\tau$ instead?
- Equation 9: the statement just before this equation states “define the transition probability as follows:” but Equation 9 is not the actual transition probability (but as you point out, it is related to the transition probability via Equation 8. It would be very helpful to discuss how the second term (with $v_\tau = 1$) is derived.
- “De novo molecular design has garnered considerable attention across various research domains in life sciences ... Among these developments, the majority of cutting-edge breakthroughs are driven by deep generative approaches ..., exemplified by Proteína ...” The callout to Proteína seems quite unnecessary, especially because it is a protein generative model, not a small molecule generative model. For proteins, the question of synthesizability is not as important as for small molecules because many proteins can be expressed in E. coli/yeast/rabbit/human cells. In the context of this paper which is focused on synthesizability, the authors should emphasize this point better.
- I feel like the first part of the title “Semi-autoregressive Your Bridge” is a little ill-formed and unclear. I would recommend the authors to modify the title, if possible.

**Ethical Concerns:**

["NO or VERY MINOR ethics concerns only"]

**Final Justification:**

Experiments with token group size were provided during the rebuttal. I was not extremely convinced because the performance goes down with increasing group size, but I think overall the methodology of the paper is well-founded and their results are strong.

**Limitations:**

yes

**Quality:**

3

**Strengths And Weaknesses:**

The paper is quite well-written, and the presentation is quite nice. Great job on the diagrams! The modelling makes a lot of sense, and the experiments seem reasonable to me.

I would have liked to see some more discussion on the chemistry aspects of MCSP; in particular, many synthesis pathways produce > 1 reaction products (eg. enantiomers), which seems hard to integrate into the current framework where the reaction output is only a single product. (One would have to represent a distribution over reaction products at each step). Another aspect is tautomerization, where the reaction product essentially exists in > 1 distinct states. Further, is there a way to add in temperature/pH dependence into your reaction templates? These could go a long way into making the approach more practically useful. For example, can your molecular fingerprints capture chirality which could help distinguish between enantiomers to choose for the next reaction?

I would have also liked to see a few more experiments varying the group size of the tokens (ie, measuring the improvement obtained by discrete diffusion vs autoregression).

---

> ### Author Rebuttal · Authors · 2025-07-30
>
> Dear reviewer,
>
> We sincerely appreciate the time you have taken to review our work and your positive comments about our paper! We aim to address your questions below.
>
> ---
>
> >**W1:** Some more discussion on the chemistry aspects of MCSP. Is there a way to add temperature/pH dependence into your reaction templates?
>
> Thank you for the insightful suggestions! We completely agree that many chemical reactions are often dependent on specific experimental conditions and additional reagents. Different experimental conditions, such as the temperature and pH you mentioned, can indeed lead to enantiomers. However, our Markov Bridge framework is highly scalable, allowing for the integration of additional constraints and prior knowledge, offering flexibility for modeling conditional dependencies, and thus can accommodate the expansion requirements for such settings. For instance, after obtaining a dataset containing condition information such as temperature and pH, the most direct approach would be to **integrate these conditions as additional context into the neural network inference process**. We believe this is a technically feasible extension. The primary limitation at present is the absence of high-quality datasets containing this kind of data. We look forward to future contributions from the community in providing data resources that include detailed reaction condition information.
>
> We believe that drawing from recent advancements in **contrastive learning techniques could also be a promising solution**. Contrastive learning works by minimizing the distance between positive sample pairs and maximizing the distance between negative pairs, allowing for the learning of more discriminative molecular embeddings, which is useful for distinguishing enantiomers that have similar structures but different chemical properties.
>
> Additionally, your observation about the influence of molecular fingerprint choice on the results is highly insightful. We believe that molecular fingerprints are essentially vector hash representations of atomic connections, local environments, paths, and other features. Regardless of the method employed, molecular fingerprints serve as encoding systems that compress a subset of the overall molecular information. We acknowledge that any molecular fingerprint systems inherently introduce some information loss. Our molecular fingerprint method iteratively updates the encoding for each atom, combining its characteristics with those of neighboring atoms, thus integrating the local environmental context of the atoms. This design allows for capturing specific substructures and local chemical environments in the molecule [1], providing molecules with similar structures whenever possible, which alleviates the issue of enantiomer differentiation to some extent. However, we also acknowledge that the current methods still require further development and optimization. Ideally, we hope that in the future, a universal encoding system can be developed that both encompasses complete molecular information and adapts to various application scenarios, including deep learning.
>
> >**W2 and Q1:** How does varying the group size affect the performance of your model?
>
> We conduct a sensitivity experiment on the group size, keeping other experimental conditions constant while changing only the group size. The results are shown in the table below. When the size is 1, the model generates tokens one by one, which is formally equivalent to an autoregressive model. However, it shows that the performance is worse compared to when the block size is 4. With a size of 6, performance slightly decreases again. This indicates that too large a size reduces the ability to model semantic dependencies, leading to limited gains. Therefore, we recommend setting the optimal size to 4.
>
> |  | Success | Recons. | Sim. (Morgan) | Sim. (Scaffold) | Sim. (Gobbi) |
> |--------|---------|---------|---------------|-----------------|--------------|
> | 1      | 0.4810  | 34.2%   | 0.7852        | 0.8139          | 0.7311       |
> | 4      | 0.4915  | 43.5%   | 0.8455        | 0.8695          | 0.8287       |
> | 6      | 0.4780  | 41.9%   | 0.8124        | 0.8313          | 0.7721       |
>
> > **Q2, Q3 and Q4:** Notations.
>
> We sincerely appreciate your thorough review of our paper and your careful examination of the mathematical details! We deeply apologize for the notation representation errors that appeared in the paper. Your understanding of the meaning of these notations is correct. We will correct all the notation issues you pointed out in the revised version. Thank you once again for bringing these issues to our attention!
>
> >**Q5:** Equation 9 is not the actual transition probability.
>
> We appreciate you pointing out this issue! Equation 9 actually defines the approximation process of the Markov Bridge. When $v_\tau$ = 1, in the Markov Bridge, the forward transfer process can be equivalently expressed as $(1-\beta_\tau)y^c+\beta_\tau t_\tau^c$. This approximation process is its reverse operation. To clarify, we will revise this claim.
>
> >**Q6:** The callout to Proteína seems quite unnecessary, especially because it is a protein generative model, not a small molecule generative model.
>
> Thank you for your valuable insight. We find your perspective very reasonable. After your suggestion, we realized that citing the protein generation model here is indeed inappropriate. We will follow your advice and remove the inappropriate reference to Proteína in the revised version, refocusing on the challenges of synthesizability in small molecule design. We will highlight the enormous waste of development resources caused by designing non-synthesizable molecules.
>
> >**Q7:** I feel like the first part of the title “Semi-autoregressive Your Bridge” is a little ill-formed and unclear.
>
> Thank you for your valuable feedback. We acknowledge that there is a lack of clarity in the title. If possible, we will remove "Semi-autoregressive Your Bridge" and revise the title according to your feedback. Once again, we sincerely thank you for your highly valuable feedback!
>
> ---
>
> Thank you again for your constructive feedback! Your feedback has been very helpful in improving our paper, and I believe you are a very responsible reviewer.
>
> Best wishes,
>
> All authors of Paper 1834.
>
> [1] Jiatong Li, Yunqing Liu, Wenqi Fan, Xiao-Yong Wei, Hui Liu, Jiliang Tang, Qing Li: Empowering Molecule Discovery for Molecule-Caption Translation With Large Language Models: A ChatGPT Perspective. IEEE Trans. Knowl. Data Eng 2024

---

> > ### Comment · Reviewer_bSFf · 2025-08-02
> > **Question about group size**
> >
> > I thank the authors for their response and their experiment on the group size. It is interesting to see that increasing the group size hurts performance; can you explain that aspect a little bit better?

---

> ### Author Response · Authors · 2025-08-02
> **Response to further question**
>
> Dear reviewer,
>
> Thanks for your further valuable feedback. Here, we provide an explanation. Considering the existence of very short synthesis pathways, larger groups limit the model's ability to capture local causal dependencies. In this case, such extensive modeling may be unnecessary for inferring these synthesis pathways. Hence, larger groups result in more potential states, which can lead to convergence difficulties, thereby affecting the model's performance.
>
> We sincerely appreciate your insightful and constructive comments, which have significantly enhanced the overall quality of our paper.
>
> Best wishes,
>
> All authors of Paper 1834.

---

> > ### Comment · Reviewer_bSFf · 2025-08-04
> >
> > Hmm, but I would have hoped that the discrete diffusion within each block is able to learn such local dependencies.
> >
> > I thank you for your responses and overall comments. I will keep my score.

---

> > > ### Author Response · Authors · 2025-08-05
> > > **Replying to Official Comment**
> > >
> > > We're grateful for your time and constructive feedback throughout the review process!! We appreciate your continued acceptance and will include your constructive suggestions in the revision.

---

### Official Review · Reviewer_xhsG · 2025-06-30

**Clarity:** 3
**Significance:** 3
**Originality:** 3
**Rating:** 5
**Confidence:** 5

**Summary:**

This work is a contribution to chemical space projection, which translates unsynthesizable molecular graph into the postfix notation-based synthesizable molecular representation. This work identified one of the key weaknesses of chemprojector: autoregressive generation is prone to error accumulation, so some error correction mechanisms are needed. Therefore, this work proposes a block-wise autoregressive generation approach. Within each block, a diffusion-like method, specifically Discrete Markov Bridge, is used to generate tokens (only reaction tokens in this work). Improved performance is demonstrated.

**Questions:**

- Only reaction tokens are modeled in the Markov Bridge. My understanding is that building block tokens have an extremely large vocabulary size (approx. 200k) and it is predicted as fingerprints, so Markov Bridge cannot be applied to fingerprints? What do the authors think the contributions of building blocks token to the error accumulation? Are errors in reactions token more detrimental? For future work, what do the authors think about the potential solution to implementing error correction on building blocks?

**Ethical Concerns:**

["NO or VERY MINOR ethics concerns only"]

**Final Justification:**

This paper is a notable technical contribution to the field and the authors mostly resolved my questions. I will keep my score.

**Limitations:**

See question and weaknesses sections

**Quality:**

3

**Strengths And Weaknesses:**

**[Strengths]**

- Chemical space projection is an important problem in molecular design because it improves the feasibility of molecules proposed by generative models. This work is a notable contribution to this field.
- The motivation is clear and sufficient. This work identified the error accumulation issue in previous work and proposed a mixed approach (specifically, block-wise autoregressive and Markov Bridge) to tackle the challenge. The mixed approach combines the strengths of both techniques. Specifically, block-wise autoregressive generation allows indefinite length sequence but lacks error correction, and Markov Bridge allows error correction but requires fixed length.
- Bottom-Up Synthesis Planning bencnmarks confirm the improvement of the proposed method.

**[Weaknesses]**

- No comparison to previous methods in other experimental settings (SBDD and Target-directed). This evaluation would be more sound if comparison of **similarity** to previous can be included in other experimental settings. It would be even better if improvement in docking/oracle scores can be shown. However, it is still okay if no significant improvement in docking/oracle is achieved because it is understandable that projection focues only on structural similarities but similarities not necessarily correlate to those scores.

---

> ### Author Rebuttal · Authors · 2025-07-30
>
> Dear reviewer,
>
> We sincerely appreciate your thoughtful and constructive feedback. Thank you for your positive assessment of our work! We now address the raised questions as follows.
>
> ---
>
> >**W:** No comparison to previous methods in other experimental settings (SBDD and Target-directed).
>
> We sincerely appreciate this valuable suggestion. However, we would like to clarify that we actually compared previous methods in the SBDD and target-directed settings, and the results included in the supplementary materials. This may have been overlooked because we did not highlight the results sufficiently in the main text, and we apologize for that. We appreciate your insightful comments. Thank you once again for your valuable time!
>
> >**Q1:** Building block tokens have an extremely large vocabulary size and it is predicted as fingerprints, so Markov Bridge cannot be applied to fingerprints? What do the authors think the contributions of building block tokens to the error accumulation? Are errors in reaction tokens more detrimental?
>
> Thank you for raising this question. We clarify that the Markov Bridge can be applied to building blocks, although there are still further improvements required in our current implementation. However, we agree that the larger vocabulary size of building block tokens indeed presents technical challenges for the direct application of the Markov Bridge. In contrast, the Markov Bridge performs better in lower-dimensional discrete spaces. For instance, it has been successfully applied in protein modeling with only 20 amino acids [1]. Of course, considering that there is still room for improvement in the discrete Markov Bridge, we also hope to further design more advanced modeling techniques. We will further consider the contributions to this specific modeling aspect.
>
> We have analyzed 7314 cases and found that 1178 cases of pathway errors were caused by building blocks, which is indeed a significant proportion. Therefore, it is also essential to address this issue. However, given the recent significant progress in guided model learning, we acknowledge that there are also better potential solutions to this issue. We will further discuss this in response to your next question (**Q2**). Thank you again for your invaluable suggestions.
>
> | | |
> |-------------|------|
> | Total cases | 7314 |
> | Errors are caused by building blocks | 1178 |
>
> >**Q2:** For future work, what do the authors think about the potential solution to implementing error correction on building blocks?
>
> To further optimize the issue of error correction in building blocks, we propose three better potential solutions.
>
> - **First**, fingerprints are essentially "encodings," which we believe are not a complete representation of the molecule but a form of information compression. If their length is too large, they even tend to cause more information loss. The Markov Bridge faces the issue of large fingerprint dimensions in its modeling process. We can draw on advanced theories from information compression to optimize its encoding process, thus generating more informative yet easier-to-handle low-dimensional representations.
>
> - **Second**, inspired by guided training methods in LLMs, we could introduce erroneous blocks as random noise in the training data, which might help the model learn erroneous patterns, potentially improving robustness to building block errors.
>
> - **Third**, akin to the second approach, during inference, we could design highly reliable reference templates for comparison to evaluate the model's performance, helping it identify modeling flaws and potentially serving as a correction strategy.
>
> Of course, these are just our preliminary technical ideas. Thank you again for your valuable professional feedback!
>
> ---
>
> Thank you once again for your thoughtful and positive review!
>
> Best wishes,
>
> All authors of Paper 1834.
>
>
> [1] Yiheng Zhu, Jialu Wu, Qiuyi Li, Jiahuan Yan, Mingze Yin, Wei Wu, Mingyang Li, Jieping Ye, Zheng Wang, Jian Wu: Bridge-IF: Learning Inverse Protein Folding with Markov Bridges. NeurIPS 2024

---

### Official Review · Reviewer_Nwry · 2025-07-03

**Clarity:** 3
**Significance:** 3
**Originality:** 3
**Rating:** 4
**Confidence:** 3

**Summary:**

The authors present MoleBridge, a method utilizing Markov bridges to semi-autogressively decode postfix notations for synthesizable chemical space projection. They compare their method to recent work in chemical space projection, particularly SynNet and ChemProjector, and find better empirical performance on a variety of tasks, including reconstruction rate, Morgan fingerprint similarity, and SBDD applications.

**Questions:**

In addition to addressing the above concerns, could the authors answer:
- Can the authors compare the empirical results with those of SynFormer (current SOTA on chemical space projection)?
- What are the computational costs associated with the Markov bridge-based method compared to the more simple autoregressive method?
- Could Figure 1 be made more clear to illustrate how the algorithm works? I find it hard to follow compared to figures demonstrating the algorithm for ChemProjector [1] and RetroBridge [2], which are both similar methods.
- I'm not sure I follow the definition of "success rate" in the evaluation. "The proportion of paths that complete at least 50% of the synthesis steps" is not clear to me. ChemProjector also uses a "success rate" but this seems to be defined differently, as the numbers are very different.

[1] Luo et al. https://arxiv.org/abs/2406.04628
[2] Igashov et al. https://arxiv.org/abs/2308.16212

**Ethical Concerns:**

["NO or VERY MINOR ethics concerns only"]

**Final Justification:**

The authors develop a novel and interesting methodology for chemical space projection, and demonstrate strong empirical results on certain benchmarks. I retain my score as though the authors addressed my questions, I am not convinced that the autoregressive formulation of prior papers is a significant issue.

**Limitations:**

yes

**Quality:**

3

**Strengths And Weaknesses:**

Strengths
- Provides an interesting and novel method to the synthesizable molecule projection task, which itself is important
- The authors demonstrate empirical benefit over ChemProjector, the most similar method explored, in many tasks such as reconstruction rate for bottom-up synthesis planning and analog generation.
- Overall, the manuscript is well-written and free of glaring errors.

Weaknesses
- The empirical benefits over ChemProjector do not seem extremely significant in many evaluations (other than reconstruction rate on the less challenging "test set"). I am curious why this evaluation does significantly better compared to other evaluations; there is something to say about the fact that this test set is more artificial and less of an extrapolative task.
- Additionally, the more recent SynFormer method [1] is not compared against, which greatly improves empirical performance compared to ChemProjector.
- The evaluations are mainly just the same ones done in ChemProjector with slightly better performance. Since the work is proposing a novel modification to the algorithm, I would be more interested in demonstrating the value behind this algorithmic change. Can the authors find ways more specific ways to show that by adding the Markov bridge / semi-autoregressive formulation, there are error modes in ChemProjector that MoleBridge addresses? For example, something like case studies where the autoregressive formulation leads to incoherent, invalid, or generally "bad" sequences that MoleBridge otherwise can correct.

[1] Gao et al. https://arxiv.org/abs/2410.03494

---

> ### Author Rebuttal · Authors · 2025-07-30
>
> Dear reviewer,
>
> We deeply appreciate your thoughtful review and your recognition of our contributions. Below, we provide point-by-point responses to your concerns and suggestions.
>
> ---
>
> >**W1:** Empirical advantage.
>
> We clarify that the test set comes from the largest and most comprehensive source of medchem building blocks in the world, and it is a standard data source. The reconstruction rate is the percentage of proposed synthesis pathways that lead to products identical to the input molecules. MoleBridge improves the error accumulation issue at the algorithmic level, leading to a higher reconstruction rate. An example can be seen in **W3**. Additionally, given the inherent difficulty of the tasks, our improvements in other evaluations have been relatively significant. In drug discovery, due to the time-consuming and labor-intensive nature of the process with low success rates [1], even a small improvement of a few percentage points could translate into millions of dollars in value. Hence, we hope that in the future, we can better demonstrate the practical application value of the algorithm. Furthermore, we hope that other excellent works in various domains will benefit from our technical ideas!
>
> >**W2 and Q1:** Can the authors compare the empirical results with those of SynFormer?
>
> Thank you for providing this paper. We observe that SynFormer is currently only published on arXiv and has not yet been peer-reviewed. Despite this, we take the suggestion seriously and have included a comparison with SynFormer.
>
> |      | Success | Recons. | Sim. (Morgan) | Sim. (Scaffold) | Sim. (Gobbi) |
> |-------------|---------|---------|---------------|-----------------|--------------|
> | SynFormer   | 0.4980  | 26.4%   | 0.7517        | 0.6827          | 0.6878       |
> | MoleBridge  | 0.4915  | 43.5%   | 0.8455        | 0.8695          | 0.8287       |
>
> Although SynFormer is an excellent parallel work, MoleBridge has advantages on most metrics. While SynFormer has a slight edge on success, we clarify that in practical studies, the primary focus is on finding synthetically viable analogs similar to the target molecule. Despite this, the comparison reveals interesting research directions, and we are exploring how to combine SynFormer’s strengths to enhance success. We will further discuss this work in the revised version.
>
> > **W3:** Case studies where the autoregressive formulation leads to incoherent, invalid sequences that MoleBridge otherwise can correct.
>
> We believe the suggestion you mentioned has unique value. To address this, we have further analyzed 7314 cases, where sequence error accumulation occurred in 5441 cases in the autoregressive formulation. We present a typical example of an early error for the target molecule, "NCCCC(=O)N(CC(=O)NN)C(N)=O". The autoregressive model generates "CC(=O)Cl" in an inference step, which is an invalid pathway, leading to the errors of subsequent steps. This is because this step cannot lead to the target structure. MoleBridge corrects the step to "NCCCC(=O)NC(N)=O", generating a complete pathway and successfully reconstructing the target molecule.
>
> >**Q2:** What are the computational costs associated with the Markov bridge-based method compared to the simpler autoregressive method?
>
> Thank you for your suggestion. With your suggestion, we have discovered an additional advantage. We conduct a computational cost comparison on the NVIDIA 4090:
>
> | Time            | Min | Average | Max   |
> |-------------------|------------|--------------|------------|
> | Autoregressive method    | 0.19s      | 10.84s       | 191.40s    |
> | MoleBridge        | 0.36s      | 9.71s        | 101.30s    |
>
> Our method is slower in inference for short sequences due to the additional overhead from the Markov Bridge. However, this slight difference in inference time is minimal, with a difference of only 0.17s, which is within an acceptable range. In contrast, for longer synthesis pathway sequences, it is faster, owing to the parallel inference of tokens within blocks.
>
> >**Q3:** Could Figure 1 be made clearer to illustrate how the algorithm works?
>
> Thank you for your revision suggestion. Since the conference reply format limits us to text only, we will follow the excellent presentation formats of ChemProjector and RetroBridge that you suggested, further simplify the process, and provide clearer and more intuitive module divisions and labeling improvements, which will be reflected in the revised version.
>
> >**Q4:** The definition of "success rate" in the evaluation.
>
> Your feedback is very important to us. We clarify that the definition of the success rate we use is the same as the one in ChemProjector, and it is computationally equivalent to the percentage of valid postfix notations, but further multiplied by 1/2, which has caused some confusion. We acknowledge that our explanation was unclear, and we will restore the original definition in the revised version.
>
> ---
> Thank you once again for your valuable and insightful review!
>
> Best wishes,
>
> All authors of Paper 1834.
>
> [1] Kang Zhang, Xin Yang, Yifei Wang, Yunfang Yu, Niu Huang, Gen Li, Xiaokun Li, Joseph C. Wu & Shengyong Yang: Artificial intelligence in drug development. Nature Medicine 2025

---

> > ### Comment · Reviewer_Nwry · 2025-08-07
> > **Reply to rebuttal**
> >
> > I thank the authors for their rebuttal. All of my questions have been addressed, but I am inclined to retain my score due to the relatively limited evaluation and improvements over prior work.

---

> > > ### Author Response · Authors · 2025-08-08
> > >
> > > **We are glad to have successfully addressed all of your questions.** We thank the reviewer for taking the time to provide feedback and maintain your acceptance score.

---

### Official Review · Reviewer_WBuU · 2025-07-03

**Clarity:** 2
**Significance:** 3
**Originality:** 3
**Rating:** 4
**Confidence:** 2

**Summary:**

This paper introduces a novel generative model, MoleBridge, for molecular rectification. It semi-autoregressive extends reaction tokens and generates postfix notations based on Markov bridges. The author validated the effectiveness of MoleBridge through a wide range of experiments.

**Questions:**

1. In the synthesis process, how accurate is the RDKit prediction of the reaction product?
2. In section 5.1, the author defines synthesis path success rate as the proportion of paths that complete at least 50% of the synthesis steps. Could the author elaborate more on why completing at least 50% of the synthesis steps is considered a success?
3. How is the number of notation blocks determined? Does the choice of the number of notation blocks affect the performance?

**Ethical Concerns:**

["NO or VERY MINOR ethics concerns only"]

**Final Justification:**

The author has addressed all my questions. I believe the work meets the standard for a NeuIPS acceptance. However, I am still not completely satisfied with the significance of the work and design of the evaluation. Therefore, I will retain my score as it is.

**Limitations:**

Yes.

**Quality:**

2

**Strengths And Weaknesses:**

**Strengths**
1. Molecule rectification is important in machine learning for molecular design. Many de novo molecule generative model generates non-synthesizable molecules. This paper uses novel methods of Markov Bridges to generate synthesis pathways.
2. The thinking and denoising separation approach in the paper successfully addresses error accumulation in chemical space projection, offering new insights into machine learning algorithms for chemical reactions.

**Weaknesses**
1. The MoleBridge generates synthetic postfix notations in the reaction pathways. The author could also discuss some earlier methods, such as JT-VAE, that also generate molecule graphs by sub-parts.
2. Could the author provide more analysis and discussion on the generated synthetic process compared to the pathway used in the real-world chemical reactions or molecular rectification?

---

> ### Author Rebuttal · Authors · 2025-07-30
>
> Dear reviewer,
>
> We are grateful for your valuable suggestions! Our detailed responses to the concerns are provided below.
>
> ---
>
> >**W1:** The author could also discuss some earlier methods, such as JT-VAE, that also generate molecule graphs by sub-parts.
>
> Thank you for sharing this paper. JT-VAE primarily involves continuous embedding and generation of molecular graphs, recursively constructing them through chemical substructures, with the aim of automating molecular design based on specific chemical properties [1]. It serves as a foundational work in the field of molecular research. Specifically, their junction tree variational autoencoder generates molecular graphs in two phases: first by creating a tree-structured scaffold over chemical substructures, and then combining them into a molecule using a graph message passing network. This is an excellent method for graph generation. We recognize the important contributions of methods such as JT-VAE in molecular structure generation.
>
> We clarify that it can generate molecular structures with specified properties, but whether these molecules can actually be synthesized through existing chemical methods remains an open question. Hence, our work tackles a **distinct yet complementary** problem, addressing downstream challenges related to its application. Thank you for your valuable suggestion. We will cite this work and discuss it. Specifically, we will further discuss the development of this field in the introduction of the revised version, providing the necessary academic discussion and recognition for important works such as JT-VAE.
>
> >**W2:** Could the author provide more analysis and discussion on the generated synthetic process compared to the pathway used in the real-world chemical reactions or molecular rectification?
>
> Thank you for this suggestion. Here, we provide additional discussions compared to the pathway used in the real-world chemical reactions from three aspects:
>
> - **First**, real-world chemical reactions typically depend on specific experimental conditions and additional reagents. Similar to most related computational methods, our model does not directly predict these conditions nor provide explicit information about the required reaction types. However, our Markov Bridge framework offers good scalability, allowing for the integration of additional constraints and prior knowledge, which offers flexibility in modeling conditional dependencies. Thus, it can accommodate the extended needs of such a setup.
>
> - **Second**, the chemical validity of our generated results primarily depends on the RDKit product prediction in the inference process. We acknowledge that this validation approach lacks direct evaluation from chemistry experts. To address this issue, we have further evaluated the accuracy of RDKit predictions (please refer to our response in **Q1**). Our analysis indicates that the prediction accuracy of RDKit is acceptable.
>
> - **Third**, we acknowledge that the reaction templates currently used are primarily derived from combinatorial chemistry libraries and may not fully represent the diversity of actual synthesis. These templates are based on validated actual reactions, ensuring chemical feasibility, and cover the most commonly used reaction types in chemistry, fulfilling the needs of most application scenarios. Additionally, future extensions and updates of the templates are also feasible.
>
> Finally, through this work, we hope to improve the design of synthesizable molecules in molecular design. As experiments in real-world scenarios are an extremely time-consuming and costly process. We believe that our contribution will provide a more reliable and practical computational foundation for chemical space projection. In the future, we hope that more excellent works in various fields will benefit from our technical ideas!
>
> >**Q1:** How accurate is the RDKit prediction of the reaction product?
>
> To evaluate the accuracy of RDKit's prediction of the reaction product, we filter all 11,818 single-step synthetic reactions generated. We evaluate the chemical validity rate (CVR) and prediction coverage (PC). CVR denotes the proportion of chemically valid predicted products, while PC indicates the proportion of samples with successfully generated predictions. As presented in the table below:
>
> | Metric | Value |
> |--------|-------|
> | CVR    | 100%  |
> | PC     | 100%  |
>
> The results indicate that the RDKit predictions are reliable.
>
> >**Q2:** Could the author elaborate more on why completing at least 50% of the synthesis steps is considered a success?
>
> Your feedback is very important to us. We have realized that there is a problem with our expression. It is computationally equivalent to the **percentage of valid postfix notations**, and further multiplied by 1/2. We acknowledge that our explanation is unclear and caused confusion, so we will restore the definition ("the percentage of valid postfix notations") and values in the revised version. We sincerely apologize and hope that this explanation clarifies the issue.
>
> >**Q3:** How is the number of notation blocks determined? Does the choice of the number of notation blocks affect the performance?
>
> We sincerely appreciate your insightful suggestion. We conduct a sensitivity experiment on the block size, keeping other experimental conditions constant while changing only the size. The results are shown in the table below. We can observe that when the size is 1, its performance is not as good as when the size is 4. When the block size is increased to 6, the performance slightly decreases again. This indicates that too large a block size reduces the ability to model semantic dependencies, leading to limited gains. Therefore, we recommend setting the optimal size to 4.
>
> |  | Success | Recons. | Sim. (Morgan) | Sim. (Scaffold) | Sim. (Gobbi) |
> |--------|---------|---------|---------------|-----------------|--------------|
> | 1      | 0.4810  | 34.2%   | 0.7852        | 0.8139          | 0.7311       |
> | 4      | 0.4915  | 43.5%   | 0.8455        | 0.8695          | 0.8287       |
> | 6      | 0.4780  | 41.9%   | 0.8124        | 0.8313          | 0.7721       |
>
> ---
>
> Thank you again for your efforts and time. We sincerely appreciate the feedback, which helped improve our work!
>
> Best wishes,
>
> All authors of Paper 1834.
>
> [1] Wengong Jin, Regina Barzilay, Tommi S. Jaakkola: Junction Tree Variational Autoencoder for Molecular Graph Generation. ICML 2018

---

> > ### Comment · Reviewer_WBuU · 2025-08-06
> >
> > Thank you for the rebuttal. The author has addressed all my questions.

---

> > > ### Author Response · Authors · 2025-08-07
> > >
> > > Thank you again for your thoughtful feedback and continued positive endorsement. We're glad we have addressed all your questions! We will carefully incorporate your suggestions into the final revision, and sincerely appreciate it if you can raise the score.

---

### Decision · Program_Chairs · 2025-09-17

**Decision:**

Accept (poster)

**Comment:**

The paper proposes a Markov Bridge generative model for postfix notations of molecular synthesis pathways. Reviewers found the work well-written, well-motivated, methodologically sound, and supported by strong empirical results. However, some expressed reservations about the significance of the contribution and the scope of evaluations (including missing baselines). These concerns were largely addressed during the rebuttal, and with all reviewers recommending acceptance, I find the contribution notable enough to merit acceptance.